# Investigation of a herpesvirus outbreak in mixed breeds of adult domestic ducks using next generation sequencing

Hassan Abu Damir[1☯], Waqar Ahmad[2☯], Neena G. Panicker[2], Layla I. Mohamed[3], Elhag A. Omer[4], Jörg Kinne[5], Ulrich Wernery[5], Abdu Adem[1,6], Mahmoud A. Ali[1]*, Farah Mustafa[2,7]*

1 Department of Pharmacology, College of Medicine & Health Sciences (CMHS), United Arab Emirates University (UAEU), Al Ain, UAE, 2 Department of Biochemistry and Molecular Biology, CMHS, UAEU, Al Ain, UAE, 3 Faculty of Science, Al Baha University, Al RAIB Al Bahah, Kingdom of Saudi Arabia, 4 Department of Agriculture, The Veterinary Laboratory, Al Ain, UAE, 5 Central Veterinary Research Laboratory (CVRL), Dubai, UAE, 6 Department of Pharmacology and Therapeutics, College of Medicine and Health Sciences, Khalifa University, Abu Dhabi, UAE, 7 Zayed Center for Health Sciences, UAE University, Al Ain, UAE

☯ These authors contributed equally to this work.
* fmustafa@uaeu.ac.ae (FM); malhaj_ali@uaeu.ac.ae (MAA)

**Data Availability Statement:** The datasets generated and/or analyzed during the current study are available in the NCBI repository, Accession no:

## Abstract

This report characterizes the first lethal outbreak of Marek's disease on a large farm of mixed-breed adult ducks (>18,000) and identifies the pathogen that resulted in high mortality (35%). Clinical signs included inappetence, respiratory distress, depression, muscle weakness, and ataxia. Post mortem revealed enlarged fragile liver mottled with miliary whitish spots and an enlarged spleen. Histopathology revealed hepatocellular necrosis with eosinophilic intra-nuclear inclusion bodies, necrosis of splenic follicles and degeneration/necrosis of renal tubules. The disease was tentatively diagnosed as a herpesvirus infection, confirmed by virus isolation from the liver. DNA was isolated from 15-year-old archival formalin-fixed tissues from infected ducks and subjected to next generation sequencing (NGS). Despite highly degraded DNA, short stretches of G- and C-rich repeats (`TTAGGG` and `TAACCC`) were identified as telomeric repeats frequently found in herpesviruses. Megablast and further investigative bioinformatics identified presence of Marek's disease virus (MDV), a Gallid alphaherpesvirus type 2 (GAHV-2), as the cause of the acute fatal infection. The source of infection may be attributed to a dead migratory flamingo found close to the duck enclosures three days prior to the outbreak; hence, GAHV-2 may also be responsible for the fatal infection of the flamingo accentuated by heat stress. Considering the possible spread of this highly contagious and lethal virus from a flamingo to the ducks, and the increasing zoonosis of animal viruses into humans, such as monkey B alphaherpesvirus transmission from macaques to humans with ~80% fatality, this observation has important ramifications for human health and safety of the poultry industry.

PRJNA859429. (https://www.ncbi.nlm.nih.gov/Traces/study/?acc=PRJNA859429&o=acc_s%3Aa).

**Funding:** This study was financially supported the United Arab Emirates University (https://www.uaeu.ac.ae/en/) in the form of grants awarded to FM (31R122, 31R140, 31M421) and AA (31M424), and NGP (31M484). This study was also financially supported by the College of Medicine & Health Sciences (CMHS) of the United Arab Emirates University in the form of a grant awarded to FM (12M092). No additional external funding was received for this study. The funders had no role in study design, data collection and analysis, decision to publish, or preparation of the manuscript.

**Competing interests:** The authors have declared that no competing interests exist.

# Introduction

The Herpesviridae family of DNA viruses has been associated with infection in a wide range of animal species and humans with variable symptoms, lesions and morbidity/mortality rates [1–3]. Based on genetic and biological similarities, herpesviridae are classified into alpha (α), beta (β), and gamma (γ) types, depending upon differences in replication rates, type of cells infected, latency sites, and cell death properties [1–3]. The α, β and γ herpesviruses cause hemorrhagic, necrotic and lytic lesions, respectively [4, 5]. Unlike the α and β herpesviridae in birds, the γ type has been reported to induce only turkey herpesvirus infection [4, 5]. The β herpesviridae has been implicated in a wide range of diseases in a larger spectrum of bird families, including Columbidae (e.g., pigeons), Psitaccidae (e.g., parrots), Falconidae (e.g., falcons), Strigidae (e.g., owls), Ciconiidae (e.g., stork), and Accipitridae (e.g., eagles), but none of these have been reported in family Anatidae (domestic and wild ducks) [1, 2, 4].

The α herpesviruses cause a variety of diseases in different species of birds, such as enteritis in ducks, infectious laryngotracheitis in gallinaceous birds, and a lymphoproliferative disease (Marek's disease) primarily in chickens, but also in quails and turkeys [1, 2, 6]. Marek's disease has been attributed to Gallid herpesvirus 2 (GAHV-2; [2, 6]. Previously classified under γ herpesviruses, it was moved to the α herpesviruses based on Marek's disease virus (MDV) genome repeat structure and sequencing [6, 7]. There are three closely related MDV serotypes: i) GAHV-2, previously known as MDV serotype 1 (MDV-1), the contagious and oncogenic strain that causes clinical disease in chickens, ii) Gallid herpesvirus 3 (GaHV-3 or MDV-2) of turkeys, and iii) meleagrid herpesvirus type 1 (MeHV-1), a turkey herpesvirus (HVT) which is non-pathogenic and has been used successfully as a vaccine against Marek's disease along with non-pathogenic GaHV-3 strains [6, 7]. GAHV-2 has also been reported in other birds in the order Galliformes; however, some like geese and ducks show no clinical signs, making them a potential reservoir of the virus for other species [8–10]. Rarely, GAHV-2 has also been reported in peafowl with fatal outcomes [11, 12]. Thus, despite host-specificity, herpesviruses are able to cross the species barrier and infect other hosts, including humans. For example, the monkey B alphaherpesvirus (endemic to Asian macaques in which the animals are mostly asymptomatic) is similar to human herpes simplex virus and can infect humans via monkey bites and scratches, causing severe encephalomyelitis and death with a mortality rate of ~80% [13, 14].

Keeping these observations in mind, the aim of this report was to describe for the first time a fatal outbreak of spontaneous infection of MDV in a mixed breed of domestic ducks, probably as a species jump from a migratory flamingo, characterize its clinicopathological features, and subsequently identify the specific viral strain involved in the outbreak. Considering that MDV has molecular and serological similarities with other lymphotropic human herpesviruses such as varicella zoster virus (VZV), herpes simplex virus (HSV), Epstein-Barr virus (EBV), and Kaposi-sarcoma associated virus (KHSV; [2, 6, 7], the potential of zoonosis into humans exists which should be kept in mind when dealing with any MDV cross-species outbreak.

# Materials and methods

## Sample collection

**Specimens sent for laboratory investigations.** Three live ducks at terminal stages and showing typical signs of disease were sent to the Central Veterinary Research Laboratory (CVRL, Dubai) for necropsy, microbiological, and parasitological investigations. Feed specimens and five ducks (three infected and two clinically normal birds) were sent to the Veterinary Diagnostic Laboratory (Qattara, Al Ain) for determination of aflatoxins in the liver and feed.

**Blood collection.** Blood samples were obtained from 49 diseased and 29 clinically normal birds (from enclosures that were not affected yet) by brachial vein into anticoagulant (EDTA) and plain vacutainers and sent to the Laboratory of the Private Department, Al Ain. EDTA blood samples were used for hematological investigations. Blood in plain vacutainers was allowed to clot and serum separated by centrifugation at 3000 RPM, kept at -20˚C, and used for biochemical analyses.

## Microbiology methods

**Virus isolation method.** The duck herpesvirus isolation was performed in the Central Veterinary Research Laboratory (CVRL), Dubai. Viral isolation was performed by tissue culture using chicken embryo fibroblasts according to routine methods.

**Bacteriological and parasitological methods.** Specimens were cultured in specific media using standard laboratory techniques for the isolation and identification of bacteria. Standard laboratory techniques were also used for parasite identification.

**Hematological & biochemical methods.** Glucose, creatinine, urea, total (T) bilirubin, total protein (TP), creatine kinase (CK), AST (aspartate aminotransferase), alanine transaminase (ALT), gamma-glutamyl transferase (GGT), and alkaline phosphatase (ALP) were analyzed in the Private Department Laboratory using Biochemistry analyzer (ACE, Alfa Wassermann, USA), and reagent kits provided by Alfa Wassermann, Netherlands. Lactate dehydrogenase (LDH) was assayed by kits from Diagnostic System (Germany). Aflatoxin B1 was estimated in feed and liver samples from infected and control birds by HPLC as per the Qattara standard laboratory protocol of specimen preparation and analysis. CBC was determined in an automated hematology analyzer (Cell-dyne 3700, USA) using hematology reagents from Abbott (Abbott Diagnostic, USA).

**Histopathological methods.** Post mortem was performed on a total of 18 birds (10 ducks were killed in extremis, 3 sent to CVRL and 5 to Qattara Lab). Representative samples from liver, spleen and kidney were obtained and fixed in 10% formol saline, processed, sectioned and stained with hematoxylin and eosin (H&E).

## Next Generation Sequencing (NGS)

**DNA extraction and quality analysis.** Whole genomic DNA was extracted from duck tissues preserved in formalin using the phenol chloroform/isoamyl alcohol method of Campos et al., 2012 [15]. DNA extraction from formalin-fixed paraffin-embedded (FFPE) samples was conducted using the ReliaPrep™ FFPE gDNA Miniprep System from Promega (Cat. No. A2351). Both the formalin-fixed duck tissues and FFPE samples were at least 15 years old when the DNA extraction was conducted. DNA quality was monitored via gel electrophoresis and amplification of the gly*ceraldehyde 3-phosphate dehydrogenase (GAPDH)* gene on a 2% agarose gel. However, all but one sample tested negative for GAPDH, revealing poor quality of DNA that was purified. To identify the viral pathogen responsible for the death of the ducks, attempts were made to PCR amplify known herpesviruses in clinical samples using a published nested PCR assay [16]. However, no amplification was observed owing to the poor quality of the formalin-treated DNA. Since we were unable to amplify any signal for either GAPDH or herpesviruses, while the nanodrop spectrophotometer readings and gel electrophoresis indicated presence of DNA, all extracted DNA samples were sent to Macrogen, Korea for commercial whole genome sequencing (WGS). Of the 12 samples sent, only two samples revealed presence of DNA in the size range of 50–200 bp using Bioanalyzer 2100 High Sensitivity chip. Due to the small size of the DNA present, library preparation was conducted without DNA shearing or size fractionation, thus bypassing the standard protocol. However, library prep

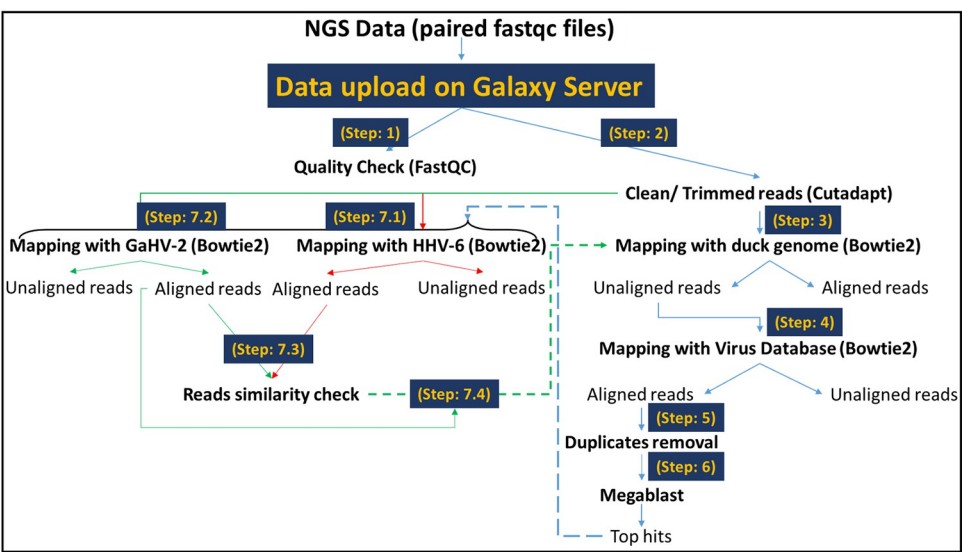

**Fig 1. Schematic diagram showing the analysis pipeline.** The analysis has been divided into 7 major steps. Galaxy server Australia was used to run the bioinformatics analysis, while IGV, IGB and NCBI Genome Workbench were used to visualize the data. Duck genome = Peking duck genome PBH1.5; GaHV-2, Gallid herpesviruses-2; HHV-6, Human herpesvirus-6.

failed with one sample due to low quantity despite using Nextera XT library, while it was successful with the other sample using the Illumina TruSeq Nano Library Kit. Therefore, the sample that passed the library quality control was subjected to NGS eventually.

**Whole genome sequencing (WGS).** WGS was performed twice on the same sample using different depths (4G & 7.5G) since results of the first analysis was not satisfactory. However, no difference was observed between results from the two depths except for the increase in number of reads in the 7.5G batch with both batches showing almost the same alignment profile either with duck reference genome or the Viral Database (C-RVDBv16.0:https://hive.biochemistry.gwu.edu/rvdb). Fig 1 represents a general outline of the data analysis pipeline followed in this study. Briefly, the raw data files generated were submitted to the Galaxy server (Australia server: https://usegalaxy.org.au/) for trimming and cleaning, followed by read mapping to the duck genome (Peking duck genome PBH1.5). The unaligned reads were mapped to the Virus Database to find presence of any viral sequences in the NGS data. Megablast was used to blast the sequences that did not align to the duck genome. The presence of human herpesvirus-6 (HHV-6) and Gallid herpesvirus-2 (GaHV-2) was further analyzed based on their hits (the two top hits), expectation values, and bit score. The genomes of both viruses were mapped to the whole NGS data as well as the unaligned reads generated while mapping with the duck genome. Prior to alignment with any of the reference genomes (duck, GaHV-2 (clone Md11BAC GenBank Accession No AY510475.1), or HHV-6 (GenBank Accession No. KY274514.2), quality check was performed on the data using FastqC which generates a report based on the overall quality of the read sequences.

A total number of 112,176,407 reads (length 151 bp) were generated after WGS using Sanger/Illumina 1.9 platform with a GC content of 41–42%. The overall quality of the data generated was acceptable (Fig 2). This was followed by adaptor removal from the raw data using Cutadapt (https://cutadapt.readthedocs.io/en/stable/). Next, the paired-end data was aligned with the available Peking duck genome PBH1.5 using Bowtie2 with default settings using Galaxy. Overall, we obtained an alignment rate of 40.84%, including all the mismatches

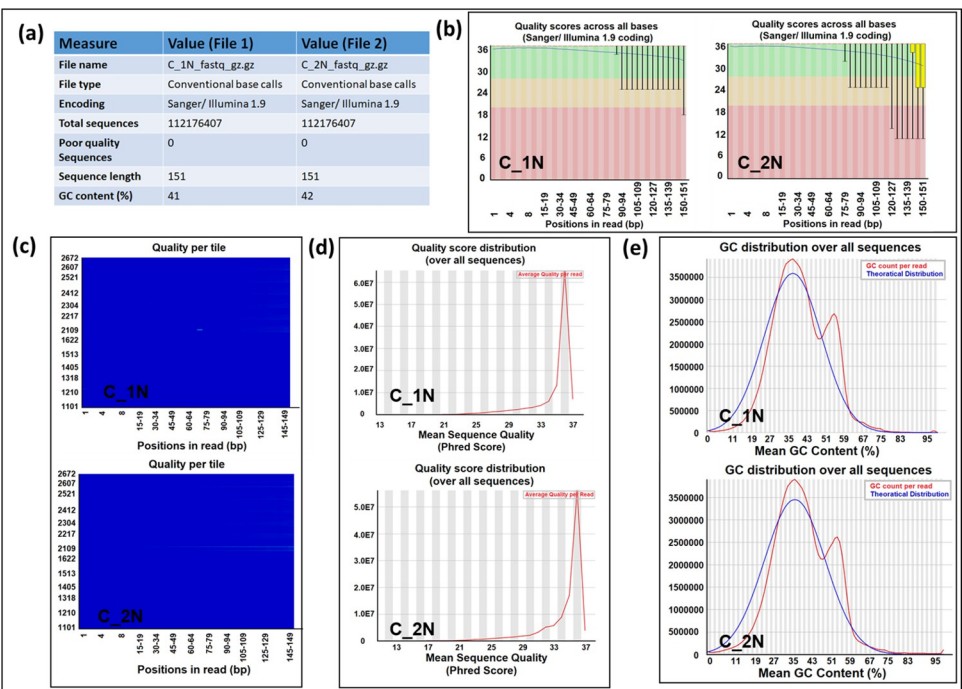

**Fig 2. Quality check of the paired data generated after sequencing.** FASTQC was used to generate the overall data quality report. (a) Basic statistics report showing the file name, type, ASCII encoding, number of total sequences, number of flagged sequences due to bad/poor quality, sequence length and % GC contents of all bases in all sequences for each file. (b) Plots showing the per base sequence quality. The whole graph shows an overview of the range of quality values across all bases at each position in the FastQ file. For each position, a BoxWhisker type plot is drawn. The central red line shows median value, the yellow box represents the inner-quartile range (25–75%), the upper and lower whiskers represent the 10% and 90% points and the blue line represents the mean quality. The y- axis shows quality score. The higher the score the better the base call. The background of the graph containing green part shows very good quality calls, orange for reasonable calls and red for poor quality calls. (c) Per tile sequence quality graph is specific for Illumina reads and shows the quality scores from each tile across all the bases to see if there was a loss in quality associated with only one part of the flow cell. The cold and hot color scheme (blue-red) represents good or bad qualities, respectively. (d) Per sequence quality score graph represents the universally low-quality values of a subset. Mean quality value > 27 shows good quality data. (e) Per sequence GC content graph represents the GC content across the length of each sequence in a file compared with a modelled normal distribution of GC content. Any shift in the normal distribution may indicate presence of some biased subsets.

and gaps, suggesting poor quality of the DNA sample used. More specifically, the aligned data showed exactly only 20.34% (22,816,353 out of 112,176,407 reads) resemblance with the duck genome. Of this, 14,603,191 reads (13.02%) reads aligned concordantly exactly once, while 8,213,162 (7.32%) aligned concordantly >1 time (more than one hit/region). The unaligned read pairs (n = 89,360,054), 13,712,736 (15.35%) aligned discordantly 1 time to the duck genome, revealing mismatches in the alignments, while the remaining 75,647,318 pairs (84.65%) aligned 0 times concordantly or discordantly. Of these 151,294,636 mates made up the pairs of which 132,729,593 (87.73%) aligned 0 times, 8,017,437 (5.30%) aligned exactly 1 time, and 10,547,606 (6.97%) aligned >1 time.

The paired-end reads were also aligned with another available duck genome with the outcome (Duckbase.refseq.v4;https://ngdc.cncb.ac.cn/gwh/Assembly/8/show). Duck has 32 chromosomes, ranging from chromosomes 1–29 and chromosomes U, W, and Z. Overall, the reads covered the whole duck genome, but regions of low/missed mapping were observed on chromosomes 12, 17, 23, 26, 27, U and W (Fig 3). This shows that the NGS data obtained was not as poor in quality as initially thought and most the duck genome could be identified in the reads obtained.

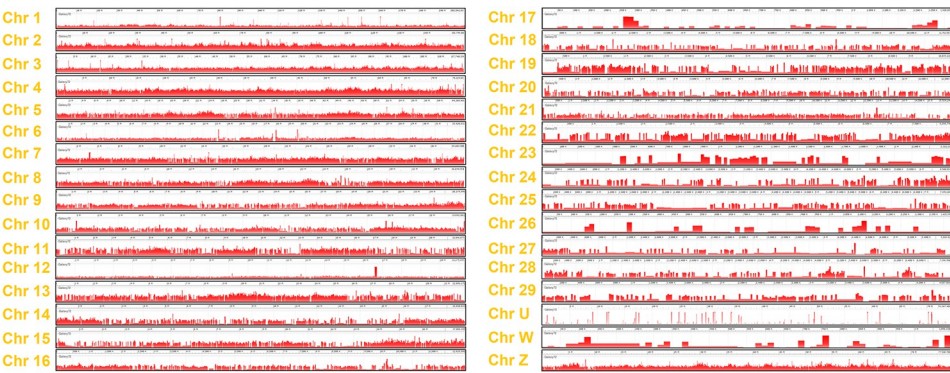

**Fig 3. Visualization of mapped reads with the duck genome PBH1.5.** After trimming, the NGS reads were aligned to the duck genome using Bowtie2. A total of 32 chromosomes have been reported in duck. NCBI Genome Workbench was used to visualized aligned reads with individual chromosome. The red lines are the histograms of the mapped reads in linear scale.

Next, the unaligned paired-end data generated was mapped to the reference Viral Database (C- RVDB v16.0) using Bowtie2 with default settings. Out of 52,490,267 paired reads, 4577 (0.008%) aligned concordantly exactly 1 time, whereas 147,338 (0.028%) aligned concordantly > 1 time. 52,338,352 (99.71%) reads remained unaligned. Thus, the overall alignment rate was 0.288% (151,915 out of 52,490,267 reads). The same outcome was generated when the raw data was directly aligned to the Viral Database where 4,577 (0.01%) reads aligned concordantly exactly 1 time and 147,338 (0.24%) aligned concordantly >1 time. The aligned FASTQ reads thus generated were converted to FASTA format using FAST-X toolkit. A total of 137,150 reads were kept, while 14,765 low quality reads were discarded by default by the program. The Dereplicate tool within Galaxy was used to remove any duplicate reads from the FASTA formatted reads to ensure there were no duplicate reads. Finally, Megablast was used to blast the reads with the Viral Database.

## Statistical analysis

Statistical analysis was conducted using SPSS (Statistical Package for Social Sciences, version 13: SPSS, Chicago, USA). Student's *t* test was used to determine significant differences between two groups. Data were expressed as mean ± SEM, and *p* value < 0.05 was considered statistically significant.

## Results

### History of the outbreak

The outbreak occurred mid-summer (July 2005) on a farm in Al Khazna, near Al Ain in the United Arab Emirates (UAE). The birds were stocked 9 months earlier and animal husbandry was under the control of a veterinarian. A total of ~18,450 adult domestic ducks of mixed French-Egyptian breeds were reared in ten big fenced enclosures about 100 meters apart. Feeding and watering were provided *ad libitum* along with lightening. The ten enclosures surrounded a big open water pool where all ducks could move around and swim freely along with other local and migrating birds during the day, while all ducks were fenced at night.

The normal mortality rate was 1–2 birds per day in each enclosure on the farm, with a total of 10–20 birds per day, and the farm never experienced any disease outbreak earlier. However, in mid-July, a migrating flamingo was found dead near the water pool around enclosure no 1.

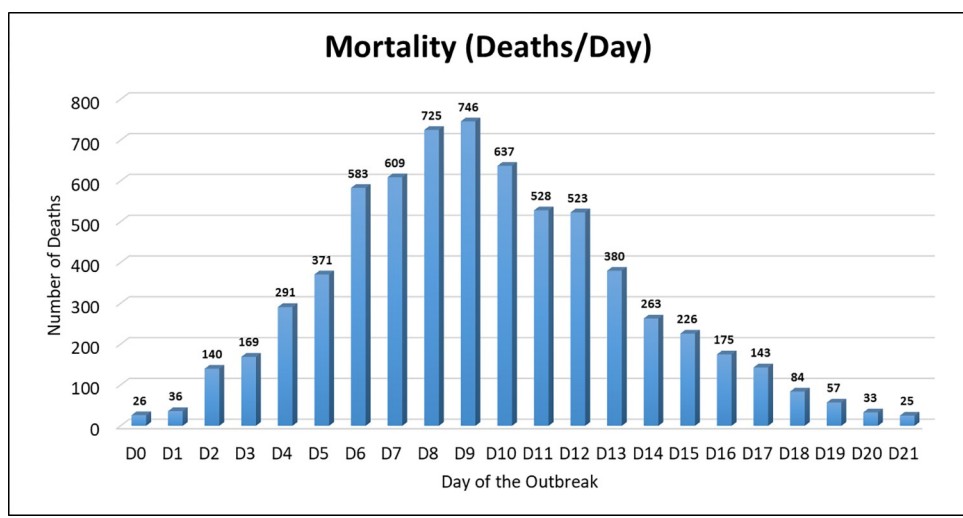

**Fig 4. Temporal analysis of duck mortality.** The number of ducks that died every day in all enclosures is mapped during the virus outbreak.

Three days later, 12, 27 and 75 birds respectively, died suddenly in enclosure no 1 on three consecutive days without showing any clinical signs of illness. Post mortem examination was performed on several birds and specimens were collected for histopathology. Consequently, the outbreak spread to different enclosures, lasting between 13–19 days in each enclosure. The preventive measures taken included: isolation of sick birds and burial of dead birds, cleaning/sanitizing of fences several times a day, and burning of excreta. Overall, a total of 6,448 died out of 18,450 ducks on the farm during the outbreak (i.e., 34.9% mortality rate) which occurred at different intervals within 45 days in every enclosure. The highest rates of death occurred between days 4–14 with peaks between days 8–9 in all enclosures where around 1,450 birds died during these two days (Fig 4).

## Clinical signs

Many ducks exhibited a peracute disease, characterized by rapid death; i.e., just found dead in the morning with no obvious clinical signs the night before. In acute cases, the ducks showed signs, including inappetence, nasal discharge, respiratory distress, watery greenish diarrhea, soiled vents, reluctance to move/swim, sluggish movement or paraparesis, muscle weakness, depression, ataxia, ruffled feather, droopy head and wings, while some males exhibited paraphimosis with abrasions.

## Post mortem findings

Necropsy was performed on 20 birds. Most of these were emaciated and displayed almost similar lesions, including enlarged fragile liver mottled with miliary whitish spots throughout the entire organ (Fig 5), hemorrhages in the mucous membrane of the gizzards, and elevated crusty plaques of diphtheritic membrane on the esophagus.

## Histopathological findings

The liver showed hepatocellular necrosis with few surrounding cells containing eosinophilic intranuclear inclusions (Fig 6A). Similarly, the pancreas showed necrotic lesions (Fig 6B). The spleen showed numerous necrotic follicles, while the kidneys of affected birds showed

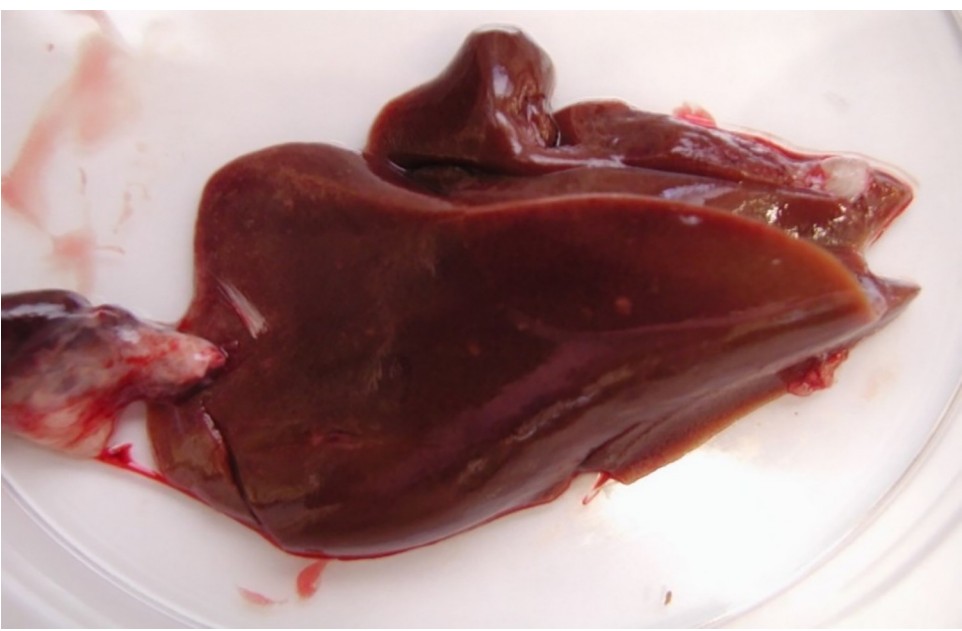

**Fig 5. Liver of a dead duck.** Image of an enlarged fragile liver from a representative duck mottled with miliary whitish spots throughout the entire organ.

degeneration and necrosis of proximal and distal convoluted tubules and collecting ducts with some shrunken glomerular tufts (data not shown).

## Microbiological findings

An unknown herpesvirus was isolated from the liver samples of three ducks by tissue culture by the clinical labs. Bacterial culture of intestinal samples was negative for salmonella, and anaerobes. However, *Acinetobacter haemolyticus* (++), *E. coli* (+) and *Mannheimia haemolytica* (*Pasteurella haemolytica*, +++) were isolated from some liver samples. The intestinal specimens were also negative for internal parasites. It is well known that the stork group of birds,

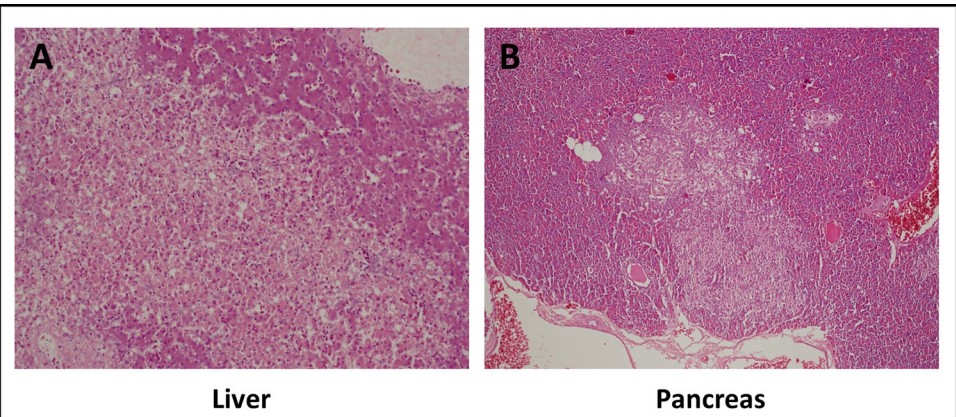

**Fig 6. Histopathological findings in infected ducks. A.** Liver of a duck. Microscopic image of a large area with hepatocellular necrosis; in contrast normal hepatic tissue in the right top area. HE, 10x objective. **(B)** Pancreas of a duck. Microscopic image of a necrotic area in the center, surrounded by normal pancreatic tissue. HE, 20x objective.

including flamingos are affected by stork inclusion body hepatitis caused by a virus of β-herpesvirinae subfamily [4]. The preliminary lab reports and liver lesions pointed to possible implication of a herpesvirus, perhaps of beta group, including causative agents of stork inclusion body hepatitis, Pacheco's disease, Pigeon inclusion body disease, Falcon herpesvirus inclusion body hepatitis, Owl herpesvirus hepatosplenitis, Crane inclusion body hepatitis, Pigeon herpes encephalomyelitis, etc., [17, 18]. To test this assumption, we checked for β-herpesviruses by target gene PCR using published primers against diverse β-herpesviruses of birds [16]. However, all DNA samples analyzed were negative for these β-herpes hepatitis viruses, necessitating the use of NGS to identify the causative agent.

## Hematological findings

Fig 7 shows the hematological results of sick and control ducks. White blood cells (WBC), lymphocytes (L) and eosinophils (E) were significantly higher ($p <$ 0.001) in sick ducks compared to the controls. However, red blood cells (RBC), hemoglobin (Hb), hematocrit (HCT), mean corpuscular volume, hemoglobin, and its concentration (MCV, MCH, & MCHC), neutrophil (N), monocytes (M), basophils (B), and platelets (PLT) were not significantly different ($p >$ 0.05) between the two groups.

## Biochemical findings

Fig 8 shows results of serum biochemical markers for the sick and control ducks. Values for the following molecules (see legends for abbreviations) were significantly higher in the serum of sick *versus* normal birds, including CK, LDH, AST, ALT, GGT, Na, K, Cl, blood urea nitrogen, total bilirubin ($p <$ 0.001) and creatinine (P $<$ 0.05), while glucose was significantly lower ($p <$ 0.05). TP concentrations were not significantly different between the two groups ($p >$ 0.05).

## Identification of the virus implicated in the outbreak using NGS

To identify the specific viral pathogen involved in this outbreak, we decided to employ the emerging technology of NGS that was less accessible and very expensive at the time of the viral outbreak. Towards this end, whole genomic DNA was extracted from duck tissues preserved in formalin that were at least 15 years old at the time of extraction (Methods). The DNA samples were sent for commercial whole genome sequencing using special library preparation kits that could use nanogram levels of DNA that was highly degraded (see Materials and Methods for details). Despite that, only one of the 12 samples sent was sufficient in quality and quantity to be sequenced. Whole genome sequencing was performed twice on the same sample using different depths (4G & 7.5G) and reads from both batches showing almost the same alignment profile either with duck reference genome or the Viral Database. The raw data was trimmed and cleaned and mapped to the Peking duck genome (PBH1.5) to remove those reads (Methods). The unaligned sequencing reads were then mapped to the Virus Database and then Megablast was used to identify the nature of the sequences that aligned to the Virus Database.

Briefly, a total of 4,329 hits were retrieved after this analysis. Table 1 summarizes the top 15 hits with $>$ 90% similarity within each hit, while S1 File provides the full dataset. Megablast results showed maximum hits with the HHV-6 (1665 hits; 38.5%) and GaHV-2 (465 + 283 = 748 hits; 17.3%). Analysis of the hits that mapped to the two viral genomes revealed that those that mapped to GaHV-2 had much better alignments and fewer mismatches/gaps than those that mapped to the HHV-6 genome, as observed by a lower E-value and higher bit score (Table 1). The alignment scores of mapped HHV-6 and GaHV-2 with NGS data were calculated using the Galaxy server tool "Qualimap 2" [19]. As can be seen, the GaHV-2

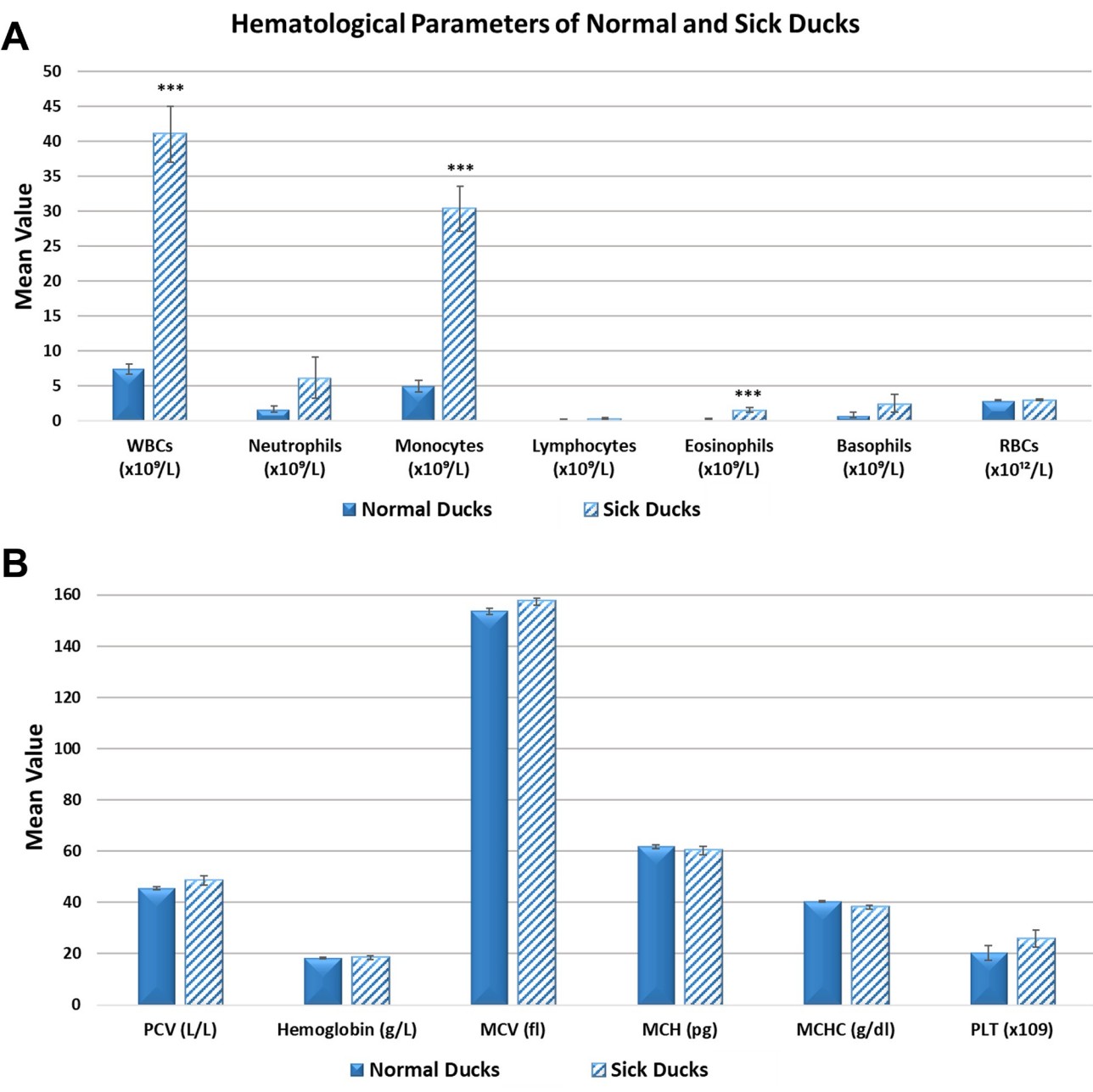

**Fig 7. Hematological parameters in infected and clinically normal ducks. Abbreviations:** WBC, white blood cells; RBC, N, neutrophils; M, monocytes; L, lymphocytes; E, eosinophils; B, basophils; RBC, red blood cells; PCV, packed-cell volume; Hb, hemoglobin; MCV, mean corpuscular volume; MCH, mean corpuscular hemoglobin; MCHC, mean corpuscular hemoglobin concentration; PLT, platelets. SEM = ± standard error of the mean. *** = $p < 0.001$.

genome showed a better mean mapping quality score of 1.17 with the NGS data with less mismatches and gaps/deletions compared to HHV-6 that had a mean mapping quality score of 0.01 (S2 File).

**Mapping of NGS reads with HHV-6 sequences.** Since our initial analysis revealed the presence of hits from HHV-6, the entire NGS data was aligned to the 161,982 bp long partial genome of HHV-6 strain HP24D3. Mapping of the NGS data with HHV-6 showed alignment of reads to three specific regions in the HHV-6 genome as follows: Region A between 8.1–8.7

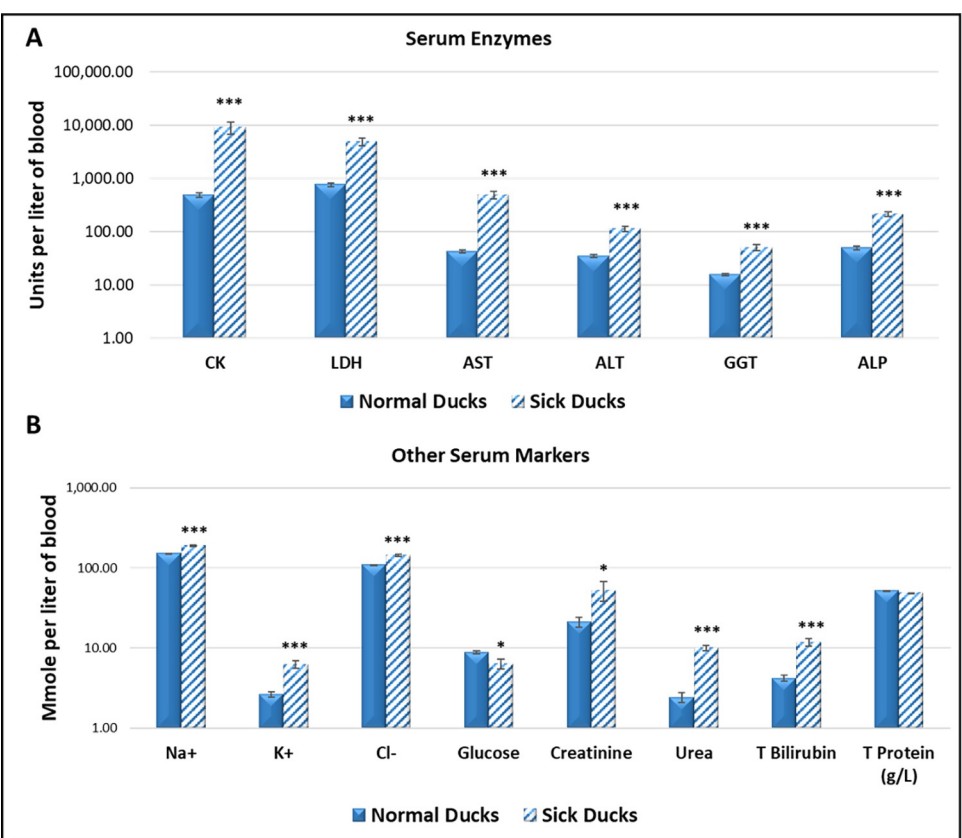

**Fig 8. Serum biochemical parameters in infected and clinically normal ducks.** Abbreviations: CK, creatine kinase; LDH, lactate dehydrogenase; AST, aspartate aminotransferase; ALT, alanine transaminase; GGT, gamma-glutamyl transferase; ALP, alkaline phosphatase; TP, total protein. SEM = ± standard error of the mean. *** = $p < 0.001$; * $p < 0.05$.

**Table 1. Top 15 hits observed after Megablast of the final filtered reads with the viral database.**

| S. No | Expectation value (E-value) | Bit score | No. of Hits | Virus Species |
|---|---|---|---|---|
| 1 | 7.76E-05 | 60.2 | 1665 | Human herpesvirus 6 strain HP24D3, partial genome |
| 2 | 5.47E-71 | 279 | 465 | Gallid herpesvirus 2 clone Md11BAC replication intermediate, complete genome |
| 3 | 7.76E-05 | 60.2 | 283 | Gallid alphaherpesvirus 2 isolate J-1, complete genome |
| 4 | 5.63E-51 | 213 | 248 | Anser cygnoides isolate B2 endogenous virus class II-related pro/pol-like gene, |
| 5 | 5.47E-71 | 279 | 218 | Homo sapiens endogenous retrovirus HERV-K, complete sequence |
| 6 | 5.67E-46 | 196 | 124 | Murine leukemia virus Graffi GV-1.2 DNA, complete genome |
| 7 | 2.62E-49 | 207 | 103 | Homo sapiens DNA, junction site of human herpesvirus 6B DNA/telomere, |
| 8 | 5.51E-66 | 263 | 96 | Xenotropic MuLV-related virus VP35, complete genome |
| 9 | 1.21E-52 | 219 | 84 | Mouse mammary tumor virus (MMTV) complete proviral genome |
| 10 | 5.63E-51 | 213 | 83 | Anser anser isolate Hui1 endogenous virus class II-related pro/pol-like gene, |
| 11 | 2.05E-40 | 178 | 42 | Cyprinid herpesvirus 1 strain NG-J1, complete genome |
| 12 | 1.19E-67 | 268 | 35 | Guanarito mammarenavirus isolate CVH-960201 segment L, complete sequence |
| 13 | 1.19E-62 | 252 | 28 | PREDICTED: Haliaeetus albicilla v-crk avian sarcoma virus CT10 oncogene homolog |
| 14 | 1.19E-62 | 252 | 23 | PREDICTED: Struthio vulgaris australis v-crk avian sarcoma virus |
| 15 | 1.19E-62 | 252 | 23 | PREDICTED: Aquila chrysaetos canadensis v-crk avian sarcoma virus CT10 oncogene |

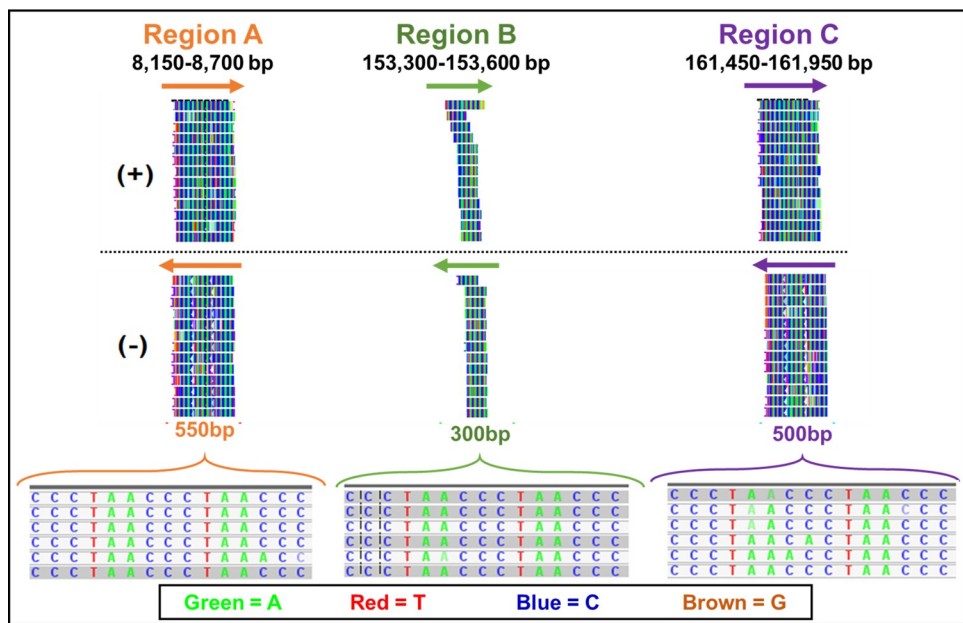

**Fig 9. Visual assessment of structural variation copy numbers after mapping of NGS data with the herpesvirus 6 genome (HHV-6) using Integrative Genomics Viewer (IGV).** Reads can be visualized as vertical lines on both strands (+/-). After mapping with HHV-6, most of the reads aligned to three regions of the viral genome that included, Region A (8150–8700 bp), Region B (153300–153600 bp) and Region C (161540–161950 bp).

kb, Region B between 153.3–153.6 kb and Region C between 161.54–161.95 kb (Fig 9). Interestingly, all three regions showed presence of repeat sequences comprising the "C" rich sequence "TAACCC" throughout. None of the NGS data mapped to any other region of the HHV-6 genome.

**Mapping of NGS reads with GaHV-2 sequences.** Next, the same read files generated after NGS were aligned to the GaHV-2 full genome (170,950 bp long) to determine if regions of homology could be observed. The results revealed that 279 reads aligned concordantly exactly 1 time, whereas 232,107 aligned concordantly >1 time (total = 279 + 232,107 = 232,386). Similar to the observation made with HHV-6, most of the reads aligned to three regions of the GaHV-2 genome: Region A corresponding to 3,650–11,150 bp, Region B corresponding to 11200–11650 bp, and Region C corresponding to 151,150–151,600 bp (Fig 10). Interestingly, similar to HHV-6, most of the reads in Region B showed a repeating pattern of "TAACCC", whereas most of the reads in Region C showed a repeating pattern of "TTAGGG" (Fig 10). However, the NGS reads mapping to Region A had equal contribution of all four bases and did not show a repeat pattern evident in Regions B and C. Furthermore, reads within Region A were distinct and did not align to any other part of the HHV-6 genome. Other than alignment with Regions A, B, and C, there were no NGS read alignments with the remaining parts of the Gallid genome either.

**Which virus infected the ducks? HHV-6 or GaHV-2.** These observations raised the question of which of these two viruses could have been the causative agent of the outbreak in ducks? To answer this question, we determined whether the reads that aligned to HHV-6 had similarity with those that aligned to the Gallid herpesvirus by mapping the HHV-6 reads to the GaHV-2 genome. Our results revealed that 99.4% of the reads (230,991 out of 232,279) aligned to the GaHV-2 genome within the same three regions shown in Fig 10. Next, pairwise alignment was performed to find any similarities between genomes of both viruses using Clustal Omega. Result of this analysis revealed an average identity and similarity of only 45.13% with average gaps of 24.39% (S3 File). As expected, only Region A of HHV-6 was observed to be

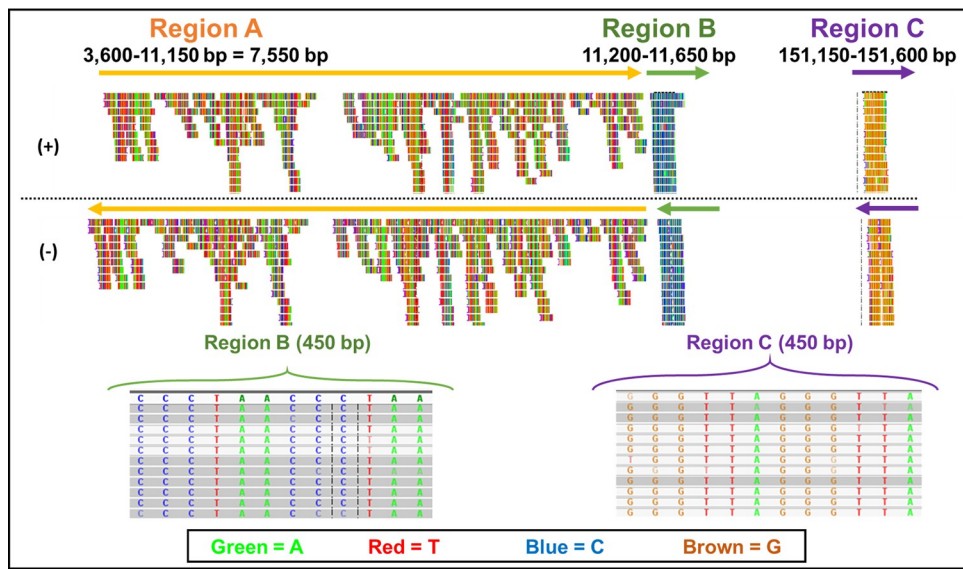

**Fig 10. Visual assessment of structural variation copy numbers after mapping of the NGS data with the Gallid herpesvirus GaHV-2 genome using the Integrative Genomics Viewer (IGV).** Reads can be visualized as vertical lines in pairs (+/-). After mapping the reads to the GaHV-2 genome, most of the reads aligned to three regions of the viral genome: Region A corresponding to 3650–11150 bp, Region B (11200–11650 bp), and Region C (151150–151600 bp).

over 96% homologous to Region B of GaHV-2 since both regions consisted of "TAACCC" (S3 File). Overall, these results suggested that the two viral genomes were quite distinct except for this region of homology which was similar only due to the similarity of repeats. Furthermore, the fact that the NGS reads observed within Regions A and C of GaHV-2 were unique and not found in HHV-6 confirms that the virus in the duck belongs to GaHV-2 or a very similar alphaherpesvirus and not HHV-6.

**Region A identified in GaHV-2 is acquired from the duck genome.** Finally, we analyzed the NGS reads that mapped to Region A in the Gallid herpesvirus genome in more detail since unlike Regions B & C, they were unique and not repeated. It has been reported in literature that Gallid herpesvirus 2 clone Md11BAC has acquired regions of the duck genome during virus evolution [20]. Therefore, we suspected that these unique reads in Region A of Gallid herpesvirus may have been acquired from the duck genome. When we mapped Region A with the duck sequences reported in GaHV-2 to be of host origin, it turns out that sequences within Region A (3,650–11,150) corresponded precisely to the integrated host duck genome sequences found within the GaHV-2 clone reported earlier (3,622–11,109 bp; [20]). This further confirms our assertion that GaHV-2 was the causative agent of the outbreak in the ducks.

## Discussion

This study reports for the first time an unusually severe outbreak of Marek's disease in adult ducks. Despite the poor quality of DNA from archival formalin-fixed tissues that were over 15 years old, NGS was able to identify the presence of GaHV-2-like sequences in the infected ducks due to the preservation of "C"- and "G"-rich repeat sequences that escaped degradation overtime [21]. A review of literature revealed that these repeat regions are actually common to many herpesviruses and have been acquired from telomeric repeat (TMR) regions of host genome by the herpesviruses over their evolutionary history [22, 23]. Even though most herpesviruses do not integrate, some herpesviruses, including α-herpesviruses, have been shown

to integrate into the host genome in regions of telomeres using these repeat sequences during latency to allow them to persist in the host permanently [23–25]. In fact, the unstable nature of the repeats and their shortening has been hypothesized as a means of virus reactivation and excision from the host genome as well [22, 23, 25].

In addition to the quality of alignment of NGS reads to GaHV-2, the presence of insertions of duck genome in the GaHV-2 genome strongly suggests GAHV-2 (MDV) as the source of the acute infection in the adult ducks as an unusual host with severe fatalities. The probability of other strains of MDV, such as GaHV-3 or MeHV-1 looks remote as these strains share only 50–80% similarity at the protein level, while GaHV-3 is non-pathogenic [6]. Interestingly, GaHV-2 is versatile in nature with frequent emergence of strains of increased virulence, most likely as a result of its ability to recombine with live attenuated vaccines introduced over the decades [2, 6, 24]. These vaccines prevent severe disease, but not infection of the vaccinated birds; thus, the infected birds can act as "carriers" of virulent strains that can further recombine to create variants with diverse disease potential and virulence [2, 6, 24, 26]. This is a hallmark of the changing clinical landscape of Marek's disease which has evolved from a more sporadic and chronic illness (polyneuritis and oncogenic) to a more aggressive and acute syndrome causing neurological symptoms, including limb paralysis and severe brain damage [2, 6]. These variants include the "mMDV" strains of moderate virulence causing inflammation, the "vMDV" virulent strains that can induce tumors, the very virulent "vvMDV" strains which induce immune suppression, and the highly virulent plus "vv+MDV" strains that induce severe immune suppression with >50% mortality [6, 27]. The disease we report seems to have been caused by a vv+ MDV GaHV-2 strain with high morbidity and a mortality rate of 35%. Since herpesviruses shed into the environment via dust particles and skin cells [28], they can spread to other animals and humans as well. However, no health risk was encountered by any of the animal handlers, veterinarians, or other farm staff during the outbreak.

The role of wild/migrating birds in this outbreak is potentially justified as a flamingo was found dead near the pool just three days before the flare up of the disease; however, post mortem was not performed as the carcass was putrefied. The MDV strain identified was probably introduced into the flock by an MDV-infected/carrier migrating bird(s), possibly the dead flamingo, through direct contact or indirectly via airborne route through inhalation of cell free virus in dust particles shed from feather dander and debris or contaminated feces and saliva [28]. Although not normally found, herpesvirus-specific sequences have been reported in captive and wild flamingos [29]. Generally, the migratory birds, including wild ducks and flamingos, are attracted to water pools inside and outside the farm and often seen together with these domestic ducks. Again, heat stress due to high ambient temperature during the month of July may be one of the factors contributing to this MDV outbreak as hot weather causes birds to flock together around water sources [30]. This is beside the fact that the virus is pervasive and environmentally stable with a long survival time [10].

Acute aflatoxicosis was ruled out in this outbreak as the cause of the outbreak as the feed samples contained 7 PPB of the toxin which was below the 20 PPB permissible limit assigned by FDA as well as liver specimens being negative for aflatoxins. Also, the liver lesions in the outbreak were different from those reported for acute aflatoxicosis in ducks and other poultry species; these liver lesions in the latter include yellowish coloration, multiple hemorrhages and characteristic reticular appearance of liver capsule [31, 32].

Hematological analysis of blood cells in the infected ducks revealed a significant rise in the absolute WBC count which was primarily due to an increase in monocytes number with slight but significant elevation in eosinophilic counts. An increase in WBC, monocytes and T cell subsets (CD4+ and CD8+) with perivascular cuffing and infiltration of monocytes in meninges has been encountered in chickens infected with the highly virulent MDV strain C12/130

without increase in B cells [27]. The RBC, PCV, Hb and blood indices of the normal ducks were comparable to that of different duck species [33, 34]. This strain of MDV seems not to cause extravascular hemolytic anemia or depression in the complete blood count (CBC) parameters similar to that reported for the highly pathogenic MDV, AC-1 and RB-1B, strains as there was no significant difference in the CBCs between the infected and normal group of ducks despite the hemorrhage, dehydration and jaundice [32, 35, 36]. It is possible that dehydration lead to hemoconcentration which masked the possible drop in CBC count by hemorrhage/hemolysis or that the death of birds was too quick for the development of anemia. Serum enzymes, AST, ALT, GGT, ALP, CK and LDH showed a significant increase in disease compared to the apparently normal ducks on the farm and to values reported for domestic ducks and other species of birds elsewhere [37]. The high activity of AST and ALT could be attributed to the generalized liver, spleen and kidney lesions [34], while the elevated GGT and ALP enzymes and bilirubinemia may be due to the liver periportal lesions in the affected birds that was fully explained in previous reports [38]. The high activity of CK and LDH in serum may be related to cardiac and/or skeletal muscle damage during the course of infection [38, 39]. The high concentration of serum Na, K, BUN and Cr may be caused by kidney lesions, whereas decreased glucose concentrations may be attributed to anorexia caused by the disease.

## Conclusions and significance

Thus, we conclude that an acute and fatal MDV infection can potentially occur in domestic ducks (and perhaps flamingos), causing mini epidemics in flocks, especially when occurring concurrently with other stress factors like heat [30]. This observation has important implications for the safety of the poultry industry that incurs losses in billions of dollars per year owing to such outbreaks [28]. More importantly, the potential of species jump into humans exists since MDV can not only recombine with other alphaherpesviruses, but also has molecular and serological similarities with several lymphotropic human herpesviruses, such as VZV, HSV, EBV, and KHSV [2, 6, 7]. Thus, MDV zoonosis should be kept in mind when dealing with any MDV cross-species transmission, as has been reported for the monkey B alphaherpesvirus outbreak from macaques to humans with ~80% fatality [13, 14] Witter, R. L. Increased virulence of Marek's disease virus field isolates. Avian Dis. 41, 149–163 (1997). Therefore, this study should expand the literature on emerging infectious diseases and how NGS can be used to address difficult questions in this field.

## Declarations

### Ethics approval

This was a field study that was initiated to investigate an acute and fatal viral infection in mixed breeds of ducks kept at a private farm near Al Ain, UAE with the full consent of the farm owner and approval of the United Arab Emirates University Animal Ethics Committee. The study involved clinical and molecular testing of blood and/or tissues samples taken from ducks that were healthy or dying. No experiments were conducted on the ducks and no protected species were sampled in this study. Details of the animal husbandry, care, and welfare have been provided in the manuscript at appropriate places.

## Supporting information

**S1 File. Full dataset of sequences of the top 15 hits shown in Table 1 after Megablast of sequences post alignment to the virus database.**
(XLSX)

**S2 File. The next generation sequence mapping quality report of human herpesvirus 6 (HHV-6) *versus* Gallid herpesvirus type 2 (GaHV-2) using the Galaxy server tool Quali-Map.**
(PDF)

**S3 File. Sequence alignment of human herpesvirus 6 (HHV-6) *versus* Gallid herpesvirus type 2 (GaHV-2).**
(PDF)

**S4 File. Highlights.**
(DOCX)

**S1 Graphical abstract.**
(TIFF)

## Author Contributions

**Conceptualization:** Hassan Abu Damir, Abdu Adem, Mahmoud A. Ali, Farah Mustafa.

**Data curation:** Hassan Abu Damir, Waqar Ahmad, Jörg Kinne, Ulrich Wernery.

**Formal analysis:** Hassan Abu Damir, Waqar Ahmad, Neena G. Panicker, Layla I. Mohamed, Elhag A. Omer, Jörg Kinne, Ulrich Wernery.

**Funding acquisition:** Abdu Adem, Farah Mustafa.

**Investigation:** Waqar Ahmad, Neena G. Panicker, Layla I. Mohamed, Elhag A. Omer, Jörg Kinne, Ulrich Wernery, Farah Mustafa.

**Methodology:** Hassan Abu Damir, Waqar Ahmad, Neena G. Panicker, Layla I. Mohamed, Elhag A. Omer, Jörg Kinne, Ulrich Wernery, Farah Mustafa.

**Project administration:** Hassan Abu Damir, Abdu Adem, Mahmoud A. Ali, Farah Mustafa.

**Resources:** Jörg Kinne, Ulrich Wernery, Abdu Adem, Farah Mustafa.

**Software:** Waqar Ahmad.

**Supervision:** Hassan Abu Damir, Abdu Adem, Mahmoud A. Ali, Farah Mustafa.

**Validation:** Waqar Ahmad, Layla I. Mohamed, Elhag A. Omer, Jörg Kinne, Ulrich Wernery, Farah Mustafa.

**Visualization:** Waqar Ahmad, Farah Mustafa.

**Writing – original draft:** Hassan Abu Damir, Waqar Ahmad, Mahmoud A. Ali, Farah Mustafa.

**Writing – review & editing:** Hassan Abu Damir, Waqar Ahmad, Abdu Adem, Mahmoud A. Ali, Farah Mustafa.

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
