## [Decision Letter · Decision Letter 0]

12 Jan 2023

Investigation of a herpesvirus outbreak in mixed breeds of adult domestic ducks using next generation sequencing

PONE-D-22-27175

Dear Dr. Mustafa,

We’re pleased to inform you that your manuscript has been judged scientifically suitable for publication and will be formally accepted for publication once it meets all outstanding technical requirements.

Kind regards,

Grzegorz Woźniakowski, Full professor, PhD, ScD

Academic Editor

PLOS ONE

Journal Requirements:

1. Please respond by return e-mail so that we can expand the acronym “UAE University” in your financial disclosure so that it states the name of your funders in full.  

We will amend your financial disclosure and competing interests on your behalf.

Reviewers' comments:

Reviewer's Responses to Questions

**Comments to the Author**

1. Is the manuscript technically sound, and do the data support the conclusions?

Reviewer #1: Yes

2. Has the statistical analysis been performed appropriately and rigorously? 

Reviewer #1: Yes

3. Have the authors made all data underlying the findings in their manuscript fully available?

Reviewer #1: Yes

4. Is the manuscript presented in an intelligible fashion and written in standard English?

Reviewer #1: Yes

5. Review Comments to the Author

Reviewer #1: This report investigated the first lethal outbreak of Marek's disease on a large farm of mixed-breed adult ducks (>18,000) with mortality considering (35%). Infected birds manifested clinical signs included inappetence, respiratory distress, depression, muscle weakness, and ataxia. During the post mortem revealed enlarged fragile liver and an enlarged spleen have been observed. DNA was isolated from 15-year-old archival formalin-fixed tissues from infected ducks and subjected to next generation sequencing (NGS). Despite highly degraded DNA, short stretches of G- and C-rich repeats (TTAGGG and TAACCC) were identified as telomeric repeats frequently found in herpesviruses.

The authors performed studies concerning: blood collection, virus isolation method concerning CEF cultures, bacteriological and parasitological studies, biochemical, histopathological and haematological analysis. Histopathology studies indicated the hepatocellular necrosis with eosinophilic intra-nuclear inclusion bodies, necrosis of splenic follicles and degeneration/necrosis of renal tubules. Authors in details conducted Nex Generation Sequencing (NGS) concerning DNA extraction and quality analysis, investigated by whole genome sequencing (WGS). The disease was tentatively diagnosed as a herpesvirus infection, confirmed by virus isolation from the liver. The statistical analysis have been performed using SPSS with Student's test which have been used to determine significant differences

The work is interesting, written in good English. I value the substantive value of the work highly. The molecular analyzes presented in it, including NGS and WGS, confirm the value of the work. The manuscript also presents a possible thread of virus interspecies transmission from flamingos, which greatly enriches the work in the epidemiological context and the assessment of the spread of MDV virus not only from another species but also from a geographical region.

The manuscript has an appropriate form and its sections fully meet the requirements of the journal. In the introduction, the reader can get acquainted with the problem, while the materials and methods section adequately describes the activities carried out, and a detailed description allows you to reproduce the work. In addition, the work contains new significant data, which is the first confirmation of MDV infection in ducks.

Due to the substantive importance of the work, I apply for acceptance of the work for publication in the journal PLOS One

6. PLOS authors have the option to publish the peer review history of their article (what does this mean?). If published, this will include your full peer review and any attached files.

Reviewer #1: **Yes: **Jowita Samanta Niczyporuk

---

## [Editor Report · Acceptance letter]

19 Jan 2023

PONE-D-22-27175 

Investigation of a herpesvirus outbreak in mixed breeds of adult domestic ducks using next generation sequencing 

Dear Dr. Mustafa:

I'm pleased to inform you that your manuscript has been deemed suitable for publication in PLOS ONE. Congratulations! Your manuscript is now with our production department. 

Kind regards, 

on behalf of

Prof. Grzegorz Woźniakowski 

Academic Editor

PLOS ONE